# Clinical manifestations and disease severity of SARS-CoV-2 infection among infants in Canada

Pierre-Philippe Piché-Renaud[1☉], Luc Panetta[2,3☉], Daniel S. Farrar[4], Charlotte Moore-Hepburn[5,6,7], Olivier Drouin[8,9], Jesse Papenburg[10,11,12], Marina I. Salvadori[10,13], Melanie Laffin[14], Fatima Kakkar[3‡*], Shaun K. Morris[1,4,7,15‡*], on behalf of the Canadian Paediatric Surveillance Program COVID-19 Study Team[¶]

1 Division of Infectious Diseases, The Hospital for Sick Children, Toronto, Canada, 2 Paediatric Emergency Department, Hospices Civils de Lyon, Hôpital Femme Mère Enfant, Lyon, France, 3 Division of Infectious Diseases, CHU Sainte-Justine, Montreal, Canada, 4 Centre for Global Child Health, The Hospital for Sick Children, Toronto, Canada, 5 Division of Paediatric Medicine, The Hospital for Sick Children, Toronto, Canada, 6 Institute of Health Policy, Management and Evaluation, University of Toronto, Toronto, Canada, 7 Department of Pediatrics, Temerty Faculty of Medicine, University of Toronto, Toronto, Canada, 8 Division of General Pediatrics, Department of Pediatrics, CHU Sainte-Justine, Montreal, Canada, 9 Department of Social and Preventive Medicine, School of Public Health, Université de Montréal, Montréal, Canada, 10 Division of Pediatric Infectious Diseases, Department of Pediatrics, The Montreal Children's Hospital, McGill University Health Centre, Montreal, Quebec, Canada, 11 Division of Medical Microbiology, Department of Clinical Laboratory Medicine, McGill University Health Centre, Montreal, Quebec, Canada, 12 Department of Epidemiology, Biostatistics and Occupational Health, McGill University, Montreal, Quebec, Canada, 13 Public Health Agency of Canada, Ottawa, Canada, 14 Canadian Paediatric Surveillance Program, Ottawa, Canada, 15 Clinical Public Health, Dalla Lana School of Public Health, University of Toronto, Toronto, Canada

☉ These authors contributed equally to this work.
‡ FK and SKM also contributed equally to this work.
¶ Membership of the Canadian Paediatric Surveillance Program COVID-19 Study Team is provided in the Acknowledgments.
* shaun.morris@mail.utoronto.ca (SKM); fatima.kakkar@umontreal.ca (FK)

**Data Availability Statement:** Data that underlie the results reported in this article (text, tables, figures and appendices) contain sensitive patient information and abide by the privacy rules of the

## Abstract

### Background

There are limited data on outcomes of SARS-CoV-2 infection among infants (<1 year of age). In the absence of approved vaccines for infants, understanding characteristics associated with hospitalization and severe disease from COVID-19 in this age group will help inform clinical management and public health interventions. The objective of this study was to describe the clinical manifestations, disease severity, and characteristics associated with hospitalization among infants infected with the initial strains of SARS-CoV-2.

### Methods

This is a national, prospective study of infants with SARS-CoV-2 from April 8th 2020 to May 31st 2021 using the infrastructure of the Canadian Paediatric Surveillance Program. Infants <1 year of age with microbiologically confirmed SARS-CoV-2 infection from both inpatients and outpatients seen in clinics and emergency departments were included. Cases were classified as either: 1) Non-hospitalized patient with SARS-CoV-2 infection; 2) COVID-19-

Canadian Paediatric Surveillance Program (CPSP) and the Public Health Agency of Canada (PHAC). De-identified data may be made available to investigators according to the processes outlined in the Privacy Act. Requests can be sent to PHAC using the email form for general inquiries under "Publications / Reports", or using the contact information detailed at the following link: https://health.canada.ca/en/public-health/corporate/contact-us.html.

**Funding:** OD & FK are supported by a Clinical Research Scholars Award, from the Fonds de recherche du Québec – Santé. We gratefully acknowledge the financial support received from the Public Health Agency of Canada to the Canadian Paediatric Surveillance Program, in support of the COVID-19 study. The funders had no role in study design, data collection and analysis, decision to publish, or preparation of the manuscript.

**Competing interests:** JP reports grants to his institution from MedImmune, Merck, Sanofi Pasteur and AbbVie, and speaker fees from AbbVie and AstraZeneca, all outside of the submitted work. MS is supported via salary awards from the BC Children's Hospital Foundation, the Canadian Child Health Clinician Scientist Program and the Michael Smith Foundation for Health Research. MS has been an investigator on projects funded by GlaxoSmithKline, Merck, Pfizer, Sanofi-Pasteur, Seqirus, Symvivo and VBI Vaccines. All funds have been paid to his institute, and he has not received any personal payments. PPPR has been co-investigator on an investigator-led project funded by Pfizer that is unrelated to this study. RP is a consultant for Verity Pharmaceuticals. SKM is co-principal investigator on an investigator-led grant from Pfizer, has served on an ad-hoc advisory boards for Pfizer and Sanofi Pasteur, and has received speaker fees from GlaxoSmithKline, all unrelated to this study. This does not alter our adherence to PLOS ONE policies on sharing data and materials.

related hospitalization; or 3) non-COVID-19-related hospitalization (e.g., incidentally detected SARS-CoV-2). Case severity was defined as asymptomatic, outpatient care, mild (inpatient care), moderate or severe disease. Multivariable logistic regression was performed to identify characteristics associated with hospitalization.

## Results

A total of 531 cases were reported, including 332 (62.5%) non-hospitalized and 199 (37.5%) hospitalized infants. Among hospitalized infants, 141 of 199 infants (70.9%) were admitted because of COVID-19-related illness, and 58 (29.1%) were admitted for reasons other than acute COVID-19. Amongst all cases with SARS-CoV-2 infection, the most common presenting symptoms included fever (66.5%), coryza (47.1%), cough (37.3%) and decreased oral intake (25.0%). In our main analysis, infants with a comorbid condition had higher odds of hospitalization compared to infants with no comorbid conditions (aOR = 4.53, 2.06–9.97), and infants <1 month had higher odds of hospitalization then infants aged 1–3 months (aOR = 3.78, 1.97–7.26). In total, 20 infants (3.8%) met criteria for severe disease.

## Conclusions

We describe one of the largest cohorts of infants with SARS-CoV-2 infection. Overall, severe COVID-19 in this age group was found to be uncommon. Comorbid conditions and younger age were associated with COVID-19-related hospitalization amongst infants.

## Background

Approximately two years into the severe acute respiratory syndrome coronavirus 2 (SARS-CoV-2) pandemic, the clinical presentation of Coronavirus Disease 2019 (COVID-19) and risk factors for disease severity among adults and older children have now been well documented [1–5]. However, there is still an important knowledge gap regarding disease manifestation and risk factors for severe disease in newborns and infants infected with SARS-CoV-2.

To date, there have been few published studies describing outcomes in infants with SARS-CoV-2 infection, with conflicting data on young age as a risk factor for severe disease [6–8]. Some studies have reported that infants are more at risk of severe disease than older children, while others have shown similar risk across pediatric age groups. More specifically, some authors have suggested that neonates (<28 days of life) [9, 10] and infants (<1 year-old) may be at higher risk of severe disease than older children [1, 11–15]. However, most of these studies have not included data on non-hospitalized and asymptomatic infants with SARS-CoV-2 infection and as such, the complete disease spectrum in this age group remains unknown.

A thorough understanding of COVID-19 in infants, including characteristics that are associated with hospitalization, would inform management and preventive interventions for this specific age group, including future vaccination strategies [16, 17]. The objective of this study was to describe the clinical manifestations and disease severity among infants with microbiologically confirmed SARS-CoV-2 infection reported to the Canadian Paediatric Surveillance Program (CPSP) between April 2020 and May 2021, and to explore infant characteristics that may be associated with hospitalization for COVID-19.

## Methods

### Study design and setting

The CPSP is a national public health surveillance platform designed to collect information on rare childhood disorders, and is administered jointly by the Canadian Paediatric Society and the Public Health Agency of Canada (PHAC) [18]. Using weekly active surveillance for incident cases and online case reporting forms, the CPSP COVID-19 study gathered information through its network of more than 2,800 paediatricians and paediatric subspecialists working in hospital and outpatient settings across Canada [18, 19]. Beginning on April 8, 2020, CPSP participants were asked to report on three different groups: (1) children admitted to hospital with acute SARS-CoV-2 infection, (2) non-hospitalized children with SARS-CoV-2 infection (symptomatic or asymptomatic) AND at least one chronic comorbid condition or less than one year of age and (3) children with multisystem inflammatory syndrome in children (MIS-C) associated with COVID-19 [20]. The detailed study protocol including case definitions and case report form for all COVID-19-related CPSP studies are available at: https://www.cpsp.cps.ca/surveillance/study-etude/covid-19. Cases could only be reported once, and duplicate data was removed.

This particular study was designed to analyze patient-level data on infants less than one year of age who were infected with SARS-CoV-2. Data on demographic characteristics, epidemiology of cases, clinical and laboratory features, treatments, and outcomes were collected. Clinical syndromes were reported directly from clinicians (e.g., pneumonia, bronchiolitis). A clinical syndrome of upper respiratory tract infection (URTI) was constructed post hoc when symptoms of coryza or rhinorrhea were reported. Cases of multisystem inflammatory syndrome in children (MIS-C) were not included in this study. To optimize case capture, this study also included additional active case finding from three large pediatric centers: The Hospital for Sick Children (HSC, Toronto, Ontario), CHU Sainte-Justine (HSJ, Montréal, Québec) and the Montreal Children's Hospital (MCH, Montréal, Québec). For these three centers, weekly extraction of infection prevention and control lists were performed using the hospital Electronic Medical Records (EMR) systems, which includes all children who had a positive SARS-CoV-2 PCR from the hospital laboratory, and all children who tested positive in another center and subsequently transferred for hospital admission. Of note, these three centers are located in the two most populous provinces that had the most COVID-19 cases in Canada during the study time frame (Québec and Ontario) [21].

### Study definitions

Cases were classified into one of the following mutually exclusive categories: 1) Non-hospitalized infants who tested positive for SARS-CoV-2; 2) COVID-19-related hospitalization: an infant hospitalized due to clinical features directly related to the SARS-CoV-2 infection; 3) Non-COVID-19-related hospitalization (e.g., incidentally detected SARS-CoV-2, isolation, or social reasons). For each hospitalized infant, at least two study team members (PPPR, LP, SKM, or FK) reviewed all available clinical and diagnostic information on the case report form to ensure the accuracy of the physician-reported case classification. Any discrepancies in the assignment of category of illness were resolved by reporting physician follow-up and/or consensus discussion by the study team.

A COVID-19 disease severity classification was adapted to the CPSP case reporting form from the World Health Organization COVID-19 Clinical Characterisation and Management Working Group scale [22], and was defined as such: 1) Asymptomatic: Patients reported to have no clinical signs or symptoms of COVID-19; 2) Outpatient care: any reported symptoms

of COVID-19 and no admission required (including from pediatric COVID-19 clinics and pediatric emergency department); 3) Mild disease (inpatient): Admitted with symptoms of COVID-19, but no targeted treatment against COVID-19 required and no respiratory support required; 4) Moderate disease: Admitted with symptoms of COVID-19 and received low-flow oxygen, had oxygen therapy above home baseline needs or received targeted treatment against COVID-19 including remdesivir or steroids; 5) Severe disease: Infants who experienced acute respiratory distress syndrome (ARDS) or severe neurological symptoms (seizures or coma), or who required intensive care unit (ICU) admission, ventilation (including high-flow nasal cannula, non-invasive ventilation, conventional mechanical ventilation, high-frequency oscillatory ventilation) or vasopressors. The detailed criteria for severity classification can be found in S1 Table. Cases were included from the first wave (occurred between March and June 2020), the second wave (September 2020 to January 2021) and the third wave (between March and May 2021) of the pandemic in Canada. With the emergence of SARS-CoV-2 variants of concerns (VOC) in Canada starting on December 26, 2020, the study was amended to gather data on SARS-CoV-2 mutation testing (N501Y and E484K) for detection of the Alpha (B.1.1.7), Beta (B.1.351) and Gamma (P.1) lineages from the three hospitals that have performed active case finding (HSC, HSJ and MCH), including on previously reported cases. Cases occurring before December 26, 2020 were assumed to be 'not a confirmed VOC'.

Infant age was categorized as neonates (<1 month; 1–28 days old), 1–3-month-old infants (29–91 days old), 4–6-month-old infants (92–182 days old) and 7–12-month-old infants (183–365 days old). These age categories were based on the significant differences in the routine management of well-appearing febrile infants aged <1 months and older infants [23]. Amongst patients aged older than 3 months, a category was created for infants aged more than 6 months of age since this population has been included in clinical trials on COVID-19 vaccination and thus have specific considerations. Prematurity was defined as gestational of less than 37 weeks, and preterm infants were included based on their chronologic age.

## Statistical analysis

Demographic characteristics were analyzed using frequencies, percentages, medians, and interquartile ranges. Differences in age groups and severity categories were assessed using Fisher's exact tests, $\chi^2$ tests, and Kruskal-Wallis tests. A statistical significance threshold of $\alpha = 0.05$ was used for all comparisons. Multivariable logistic regression was performed to identify infant characteristics associated with hospitalization due to COVID-19, after excluding infants hospitalized with incidental SARS-CoV-2 infection. Variables for the primary adjusted model were selected a priori on the basis of literature review and available elements on the case report form. Given a smaller available sample size for the post hoc VOC model, the variable with the lowest pairwise correlation with hospitalization (i.e., infant sex) was excluded while VOC status was forced into the adjusted model. Odds ratios (OR) and 95% confidence intervals (CI) were calculated. Unadjusted comparisons between infants hospitalized because of COVID-19 with severe and non-severe disease were performed using a Wilcoxon rank-sum test and chi-square test (excluding cases that were admitted for reasons other than COVID-19). Logistic regression to identify factors associated with disease severity could not be performed due to the small number of infants with severe disease. Sensitivity analyses were conducted using only the cases from the three centres that systematically screened for inclusion all infants who tested positive for SARS-CoV-2 (HSC, HSJ and MCH). Analyses were conducted using Stata 17 (StataCorp, College Station, TX). Due to CPSP privacy policies, frequencies between one and four were masked and reported as '<5'. In some instances, the results of subgroup analyses were presented as ranges to prevent back-calculation of frequencies <5.

### Ethical approval

The CPSP operates under the legal authority derived from Section 4 of the Department of Health Act and Section 3 of the Public Health Agency of Canada Act. Ethics approval was obtained from the Public Health Agency of Canada and Health Canada Research Ethics Board (REB 2020-002P), The Hospital for Sick Children (REB 1000070001) and the Institutional Review Board of the Centre Hospitalier Universitaire Sainte-Justine (IRB MP-21-2021-2901) as well as at individual sites, as required by local policies. In Québec, the study was conducted as a multicenter study, with clinical data collected by the study co-investigators. Operating under contract with PHAC, the CPSP, a program of the Canadian Paediatric Society (CPS), has the authority to collect health information for surveillance purposes without patient consent. Data was reported and collected anonymously.

## Results

### Patient demographics

From April 8[th], 2020 to May 31[st], 2021 a total of 531 infants aged <1 year old were reported to CPSP, including 199 (37.5%) hospitalized and 332 (62.5%) non-hospitalized infants (Fig 1). Cases were reported from all provinces with the majority from Québec and Ontario (Fig 2). Full data capture was not possible for Alberta because of delays in securing the appropriate data-transfer agreements. Eighty-one infants (15.4%) were neonates (≤28 days old), of whom 13 were ≤7 days old and 67 between 8 to 28 days old (one infant could not be categorized). Most infants had a known close contact with a confirmed case of SARS-CoV-2 infection (n = 302, 58.5%), with either their parents (n = 247; 79.2%), siblings (n = 46, 14.7%), other relatives (n = 51, 16.4%), or non-family members (n = 13, 4.2%). Other infant characteristics are shown in Table 1.

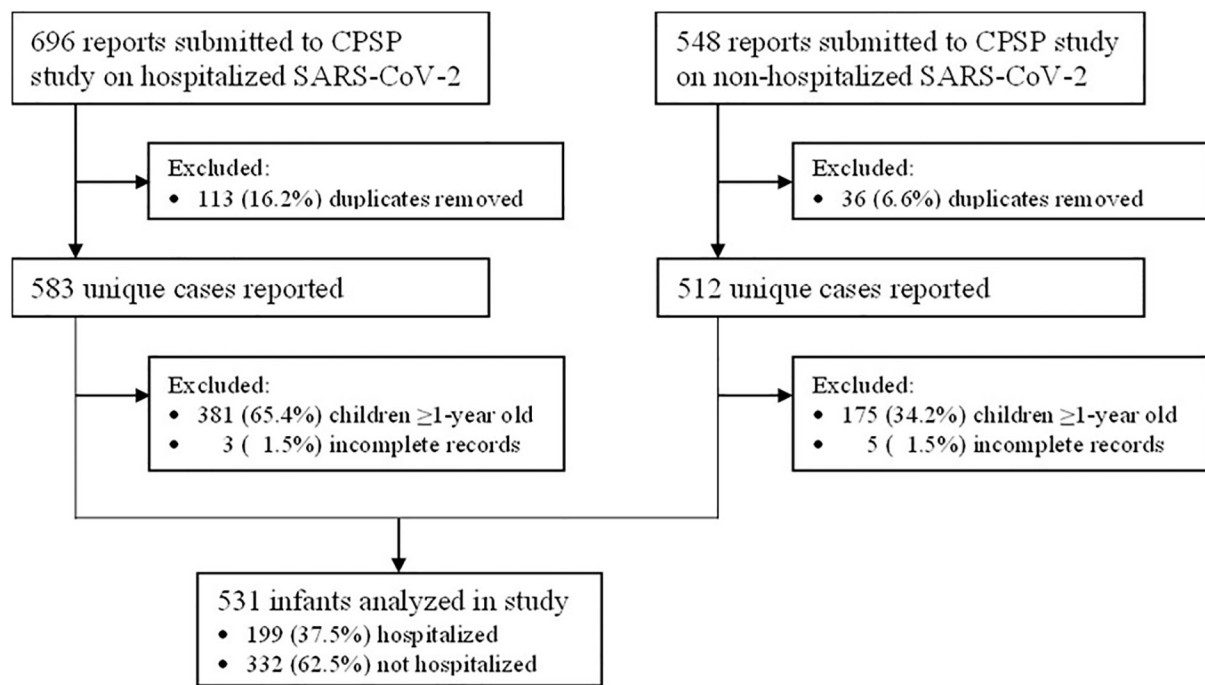

**Fig 1. Flow chart of cases reported to the CPSP COVID-19 study until May 31, 2021.**

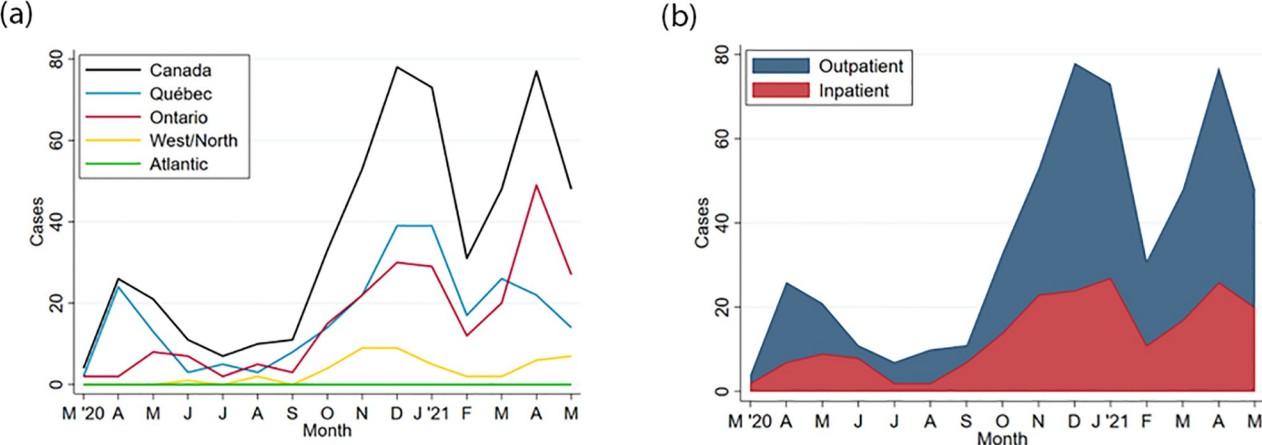

**Fig 2. a.** Cases of SARS-CoV-2 infection among infants reported to the CPSP, by Canadian region*. *Western/Northern Canada includes Alberta, British Columbia, Manitoba, Northwest Territories, Nunavut, Saskatchewan, and Yukon. Atlantic Canada includes Newfoundland & Labrador, New Brunswick, Nova Scotia, and Prince Edward Island. **b.** Cases of SARS-CoV-2 infection among infants reported to the CPSP, by admission status.

## Clinical characteristics

From all reported infants, 66 (12.4%) were asymptomatic from SARS-CoV-2 infection (Table 2). Among symptomatic infants, the most common presenting symptoms were fever (n = 353, 66.5%), coryza (n = 250, 47.1%), cough (n = 198, 37.3%) and decreased oral intake (n = 133, 25.0%) (Table 2). The most common clinical syndromes were URTI (n = 321, 60.5%), followed by gastrointestinal syndrome (161, 30.3%). Few infants had bronchiolitis (n = 20, 3.8%) or pneumonia (n = 14, 2.6%). A total of 59 infants (11.1%) had at least one comorbidity, including congenital heart disease (n = 20, 3.8%), neurologic/neurodevelopmental disease (n = 19, 3.6%) and chronic lung disease (n = 10, 1.9%). Fourteen infants (2.6%) had more than one co-morbidity. Most infants were born at term (n = 445, 88.8%), with 29 (5.8%) born prematurely between 34 and 36 weeks of gestation and 27 infants (5.4%) born before 34 weeks of gestation (Table 1).

## Hospitalization for COVID-19

Among hospitalized infants, 141 of 199 infants (70.9%) were admitted because of COVID-19-related illness, and 58 (29.1%) were admitted for reasons other than acute COVID-19 (Table 1). Neonates (<1 month) were more likely to be hospitalized for COVID-19, with an adjusted odds ratio (aOR) of 3.78 (95% CI, 1.97–7.26) when compared with infants aged 1–3 months. Infants aged 4–6 months and 7–12 months were less likely to be admitted than infants aged 1–3 months with an aOR of 0.24 (95% CI, 0.12–0.47) and aOR of 0.13 (95% CI, 0.06–0.25), respectively (Table 2). Amongst infants aged less than one month of age who were admitted, 36 (73.5%) experienced fever, which likely influenced the decision for these infants to be hospitalized (S2 Table). The median age for infants managed as outpatients was 5.3 months (Interquartile range [IQR], 2.4–8.6 months), and the median age for infants admitted for COVID-19-related illness was 1.2 months (IQR, 0.7–3.8 months, p<0.001). Odds ratios for COVID-19 admission by continuous infant age, with infants at 3.8 months (100 days) of age as the reference point are reported in the S1 Fig. Infants born before 34 weeks of gestation were at increased odds of being hospitalized for COVID-19 compared to infants born at term (aOR, 3.54; 95% CI, 1.19–10.46). Infants with ≥1 comorbid condition(s) were also at increased odds

**Table 1. Characteristics of infants with SARS-CoV-2 infection in Canada.**

| Characteristics | All Infants | Outpatients | Inpatients | | P value |
|---|---|---|---|---|---|
| | | | Not COVID-19 related | COVID-19 related | |
| **Total cases reported, N** | 531 | 332 | 58 | 141 | - - - |
| **Infant age (months), median (IQR)[1]** | 3.7 (1.4–7.9) | 5.2 (2.5–8.6) | 4.5 (1.2–8.6) | 1.2 (0.7–2.9) | <0.001 |
| **Infant age category, n (%)[1]** | | | | | <0.001 |
| <1 month | 81 (15.4) | 20 (6.1) | 12 (21.4) | 49 (35.0) | - - - |
| 1–3 months | 151 (28.8) | 82 (24.9) | 13 (23.2) | 56 (40.0) | - - - |
| 4–6 months | 112 (21.3) | 81 (24.6) | 13 (23.2) | 18 (12.9) | - - - |
| 7–12 months | 181 (34.5) | 146 (44.4) | 18 (32.1) | 17 (12.1) | - - - |
| **Infant sex, n (%)** | | | | | 0.45 |
| Female | 232 (43.7) | 152 (45.8) | 23 (39.7) | 57 (40.4) | - - - |
| Male | 299 (56.3) | 180 (54.2) | 35 (60.3) | 84 (59.6) | - - - |
| **Population group of infant, n (%)[2]** | | | | | - - - |
| White | 90 (16.9) | 32 (9.6) | 13 (22.4) | 45 (31.9) | <0.001 |
| South Asian | 63 (11.9) | 38 (11.4) | 6 (10.3) | 19 (13.5) | 0.77 |
| Arab/West Asian | 41 (7.7) | 15 (4.5) | 5 (8.6) | 21 (14.9) | <0.001 |
| Black | 40 (7.5) | 19 (5.7) | 8 (13.8) | 13 (9.2) | 0.06 |
| East/Southeast Asian | 25 (4.7) | 18 (5.4) | <5 (<8.6) | <5 (<3.5) | 0.48 |
| Indigenous | 16 (3.0) | <5 (<1.5) | 6–9 (10.3–15.5) | 5–8 (3.5–5.7) | <0.001 |
| Other | 14 (2.6) | 9–12 (2.7–3.6) | <5 (<8.6) | <5 (<3.5) | 0.57 |
| Unknown | 254 (47.8) | 204 (61.4) | 15 (25.9) | 35 (24.8) | <0.001 |
| **Region of residence, n (%)[3]** | | | | | - - - |
| Québec | 251 (47.3) | 162 (48.8) | 21 (36.2) | 68 (48.2) | - - - |
| Ontario | 233 (43.9) | 164 (49.4) | 24 (41.4) | 45 (31.9) | - - - |
| West/North | 47 (8.9) | 6 (1.8) | 13 (22.4) | 28 (19.9) | - - - |
| Atlantic | 0 (0.0) | 0 (0.0) | 0 (0.0) | 0 (0.0) | - - - |
| **Gestational age at birth, n (%)[4]** | | | | | <0.001 |
| Term (≥37 weeks') | 445 (88.8) | 284 (93.1) | 40 (70.2) | 121 (87.1) | - - - |
| Late preterm (34–<37 weeks') | 29 (5.8) | 12 (3.9) | 9 (15.8) | 8 (5.8) | - - - |
| Moderate/very preterm (<34 weeks') | 27 (5.4) | 9 (3.0) | 8 (14.0) | 10 (7.2) | - - - |
| Median (IQR) weeks at birth[5] | 34.0 (31.4–35.5) | 34.0 (32.4–35.7) | 34.6 (30.0–35.0) | 32.9 (29.0–35.4) | 0.48 |
| **Any comorbid condition, n (%)** | 59 (11.1) | 23 (6.9) | 18 (31.0) | 18 (12.8) | <0.001 |
| Congenital heart disease | 20 (3.8) | 9 (2.7) | 6 (10.3) | 5 (3.5) | 0.03 |
| Neurologic/neurodevelopmental | 19 (3.6) | 6–9 (1.8–2.7) | 8–11 (13.8–19.0) | <5 (<3.5) | <0.001 |
| Chronic lung disease | 10 (1.9) | <5 (<1.5) | <5 (<8.6) | 5 (3.6) | 0.007 |
| Other[6] | 25 (4.7) | 325 (97.9) | 49 (84.5) | 132 (93.6) | <0.001 |
| **Any co-infections, n (%)[7]** | 47 (8.9) | 17 (5.1) | 18 (31.0) | 12 (8.5) | |
| **COVID-19 exposure, n (%)** | | | | | - - - |
| Known close contact with confirmed SARS-CoV-2 infection | 312 (58.8) | 187 (56.3) | 28 (48.3) | 97 (68.8) | 0.01 |

(*Continued*)

**Table 1.** (Continued)

| Characteristics | All Infants | Outpatients | Inpatients | | P value |
| --- | --- | --- | --- | --- | --- |
| | | | Not COVID-19 related | COVID-19 related | |
| Nosocomial infection | 9 (1.7) | 0 (0.0) | 5–8 (8.6–13.8) | <5 (<3.5) | <0.001 |

[1]Continuous age not available for 13 infants (8 outpatients, 2 non-COVID admissions, 3 COVID admissions); categorical age not available for 6 infants (3 outpatients, 2 non-COVID admissions, 1 COVID admission).

[2]Physicians could report multiple population groups. East/Southeast Asian includes Chinese, Filipino, Japanese, Korean, and Southeast Asian. Indigenous includes First Nations, Inuit, and Métis.

[3]Overall, West/North includes cases from Manitoba (n = 16), Alberta (n = 14), Saskatchewan (n = 10), and either Northwest Territories, Nunavut, or British Columbia (n = 7). Zero cases were reported from Yukon. Atlantic includes New Brunswick, Newfoundland and Labrador, Nova Scotia, and Prince Edward Island.

[4]Gestational age category not known for 30 infants (27 outpatients, 1 non-COVID admission, 2 COVID admissions).

[5]Among preterm-born infants only (i.e. <37 weeks' gestation).

[6]Includes 8 gastrointestinal/liver disease, 8 sickle cell/other hematologic disorder, 6 immunosuppressing conditions, 5 renal disease, and <5 not otherwise specified. Multiple conditions may have been reported.

[7]Includes 19 urinary tract infections, 7 invasive bacterial infections, 6 concomittant viral infections, 5 acute otitis media/bacterial pneumonia, and <5 each of candidiasis, laryngitis/pharyngitis, other benign bacterial infections, and skin and soft tissue infections.

of hospitalization for COVID-19 compared to infants with no known comorbid conditions (aOR, 4.53; 95% CI, 2.06–9.97). Sensitivity analyses were conducted to confirm results among the two Montreal and the Toronto centres with systematic case reporting. While associations between hospitalization, comorbidities, and young age remained consistent, these analyses showed a higher proportion of non-hospitalized infants (n = 275/374, 73.5%) and the association between preterm birth before 34 weeks of gestation and hospital admission was not statistically significant (aOR 2.21; 95% CI, 0.55–8.95, S3 and S4 Tables). In these analyses, similarly to the analyses performed on the full dataset, comorbidities and young age were also associated with admission for COVID-19 (S2 Table).

## Severe COVID-19 disease amongst infants hospitalized for COVID-19

Amongst infants hospitalized because of COVID-19-related illness, 111 of 141 infants (78.7%) had mild disease, 10 (7.1%) had moderate disease and 20 (14.2%) had severe disease. A comparison of disease severity classification of our cases using the objective CPSP criteria, versus the Dong severity criteria is shown in S5 Table. Fewer than five infants with COVID-19 died. Fourteen infants (9.9%) were admitted to the ICU, with <5 infants requiring non-invasive ventilation and 5 infants requiring mechanical ventilation (S6 Table). A higher proportion of neonates aged <1 month (26.5%) admitted for COVID-19 required respiratory support (oxygen, non-invasive ventilation and mechanical ventilation) compared to infants aged 1–12 months (11.0%, p = 0.02) (S2 Table). The median age of inpatients with severe disease was 0.6 months (IQR 0.3–2.7) compared to 1.4 months (IQR 0.8–3.1) among inpatients with non-severe disease (p = 0.03). Inpatients with severe disease were more likely to be premature (<37 weeks' gestation; 7/20 [35.0%]) than inpatients with non-severe disease (11/119 with known prematurity status [9.2%]; p = 0.005).

## Treatment and management

Almost all infants managed as outpatients (314 of 332, 94.6%) did not receive any treatment, whereas 61 (43.3%) infants hospitalized for COVID-19 did not receive any treatment (p<0.001) (S7 Table). Antibiotics were the most frequent treatment modality (94/473, 19.9%),

**Table 2. Logistic regression analysis of infants' characteristics associated with COVID-19 hospital admission.**

| Characteristics, n (%row) | Crude model | | Adjusted model[1] | | VOC adjusted model[2] | |
|---|---|---|---|---|---|---|
| | OR (95% CI) | *P* value | aOR (95% CI) | *P* value | aOR (95% CI) | *P* value |
| **Infant age** | | | | | | |
| 0–<1 month | 3.59 (1.93–6.68) | <0.001 | 3.78 (1.97–7.26) | <0.001 | 3.47 (1.42–8.52) | 0.007 |
| 1–3 months | 1 [Reference] | NA | 1 [Reference] | NA | 1 [Reference] | NA |
| 4–6 months | 0.33 (0.18–0.60) | <0.001 | 0.24 (0.12–0.47) | <0.001 | 0.30 (0.12–0.78) | 0.01 |
| 7–12 months | 0.17 (0.09–0.31) | <0.001 | 0.13 (0.06–0.25) | <0.001 | 0.12 (0.05–0.34) | <0.001 |
| **Infant sex** | | | | | | |
| Female | 1 [Reference] | NA | 1 [Reference] | NA | - - - | - - - |
| Male | 1.24 (0.83–1.86) | 0.28 | 1.14 (0.71–1.84) | 0.58 | - - - | - - - |
| **Gestational age at birth** | | | | | | |
| Term (≥37 weeks') | 1 [Reference] | NA | 1 [Reference] | NA | - - - | - - - |
| Late preterm (34–<37 weeks') | 1.56 (0.62–3.92) | 0.34 | 1.25 (0.42–3.72) | 0.69 | - - - | - - - |
| Moderate/very preterm (<34 weeks') | 2.61 (1.03–6.58) | 0.04 | 3.54 (1.19–10.46) | 0.02 | - - - | - - - |
| **Comorbid conditions** | | | | | | |
| None/Unknown | 1 [Reference] | NA | 1 [Reference] | NA | 1 [Reference] | NA |
| ≥1 comorbid condition | 1.97 (1.03–3.77) | 0.04 | 4.53 (2.06–9.97) | <0.001 | 7.98 (2.74–23.29) | <0.001 |
| **Phase of COVID-19 pandemic** | | | | | | |
| 1st wave (April–August 2020) | 1 [Reference] | NA | 1 [Reference] | NA | - - - | - - - |
| 2nd wave (September 2020–February 2021) | 1.20 (0.65–2.19) | 0.56 | 1.37 (0.68–2.75) | 0.38 | - - - | - - - |
| 3rd wave (March–May 2021) | 1.16 (0.61–2.20) | 0.64 | 1.57 (0.74–3.31) | 0.24 | - - - | - - - |
| **Variant of concern** | | | | | | |
| Not a confirmed VOC[3] | 1 [Reference] | NA | - - - | - - - | 1 [Reference] | NA |
| Confirmed VOC | 0.81 (0.37–1.76) | 0.59 | - - - | - - - | 0.63 (0.27–1.48) | 0.29 |

aOR = Adjusted odds ratio; OR = Odds ratio; VOC = variant of concern.

[1]Multivariable analysis was conducted among 440 complete cases, and excludes 58 non-COVID-19-related hospitalizations.

[2]Multivariable analysis conducted among 268 complete cases reported from the Centre Hospitalier Universitaire Sainte-Justine, Hospital for Sick Children, or Montreal Children's Hospital and occurring on or after September 1, 2020. Excludes 25 patients hospitalized for reasons other than COVID-19. Given a smaller available sample size, VOC status was forced into the multivariable model and other covariates were selected on the basis of p-values until an events-per-variable ratio of 1:10 was reached to avoid overfitting.

[3]Cases occurring before December 26, 2020 (i.e. first case of confirmed Alpha in Canada) were assumed as 'not a confirmed VOC'. Cases with no VOC screening and occurring on or after December 26, 2020 (n = 119) were also assumed as 'not a confirmed VOC'. Excluding the latter cases from the multivariable analysis did not meaningfully change the aOR (0.62; 95% CI 0.24–1.66).

and neonates aged <1 month received antibiotics more frequently (35/69, 47.7%) than infants aged 1–3 months (34/138, 24.6%) and infants aged 4–12 months (25/262, 9.5%, p-<0.001). Few infants received systemic corticosteroids (11/473, 2.3%), including seven infants who received steroids for a non-COVID-19 related reason (for example, laryngitis). Less than five infants (<1.1%) received remdesivir, and no patients were treated with biologic response modifying drugs, anticoagulation (e.g. prophylactic, therapeutic, mechanical), or chloroquines (e.g. chloroquine, hydroxychloroquine).

## SARS-CoV-2 variants of concern

From centers contributing SARS-CoV-2 lineage data (HSC, HSJ and MCH), 60 infants screened positive for VOC, all of whom tested positive on or after February 24, 2021. This included 39 infants with the Alpha VOC and 21 infants with another VOC (including Beta,

Gamma or unspecified N501Y+ VOC) (S4 Table). In our VOC adjusted model, no SARS-CoV-2 lineage was associated with an increased odds of hospitalization in infants (Table 3). The aOR of hospitalization for VOC compared to the SARS-CoV-2 ancestor strain was 0.63 (95% CI, 0.27–1.48). The phase of the pandemic (first, second or third wave) was not associated with an increased odds of hospitalization in infants.

## Discussion

This study describes the clinical manifestations and severity of disease in infants less than one year of age with microbiologically-confirmed SARS-CoV-2 infection in Canada in outpatient and hospital settings prior to May 2021. While previous studies on COVID-19 in infants have focused only on hospitalized patients with symptomatic disease and may therefore have over-estimated the severity of COVID-19 in this age group, this study provides a more complete description of the possible range of presentations of infants with SARS-CoV-2 infection by including a subset of infants seen in outpatient settings [9, 24, 25].

Similar to the results of previous studies, including meta-analyses and systematic reviews on pediatric COVID-19, we report that severe COVID-19 is uncommon in infants, especially when compared to adults and adolescents [26–29]. It is thought that the difference in severity from SARS-CoV-2 infection between age groups relates to distinct immune responses and expression in receptors of angiotensin-converting enzyme-2 (ACE-2) in younger children [30]. However, consistent with previous studies, we found that amongst infants, neonates aged <1 month were at increased risk for hospitalization [9, 10, 13, 15, 31]. Although a low threshold for admission in infants aged less than 6 to 12 weeks who present with fever may impact hospitalization rates, a larger proportion of neonates aged <1 month admitted for COVID-19 have also required respiratory support compared to older infants, including 5 who required mechanical ventilation [23]. The increased severity of SARS-CoV-2 infection seen in young neonates may relate to their small airways and reduced innate immune responses, especially in the absence of passive immunity from maternal antibodies against SARS-CoV-2 [32, 33]. Several studies and reports have demonstrated successful placental transmission of SARS-CoV-2 antibodies after maternal vaccination [34–36]. Thus, in the absence of any COVID-19 vaccine approved for use in young infants, an important strategy to protect neonates against severe SARS-CoV-2 may be through maternal immunization. With multiple jurisdictions reporting lower COVID-19 vaccine coverage in pregnant women compared to the general population, efforts should be made to optimize their acceptance and uptake of COVID-19 vaccines [37, 38].

Our results also suggest that preterm infants (<37 weeks of gestation) may be at higher risk of severe COVID-19 compared to term infants. While a multivariable analysis revealed higher odds of admission in preterm infants, this result was not consistent in our sensitivity analysis on the data from centres with systematic case reporting. To date, there have been very few data describing the spectrum of COVID-19 manifestations in infants who are born prematurely. Some case reports have described severe disease requiring intubation in premature neonates infected with SARS-CoV-2, whereas others have described an overall mild course [39–43]. Thus, further research on the spectrum of illness of SARS-CoV-2 infection in this population is needed.

The emergence of new SARS-CoV-2 lineages have raised concerns for differences in susceptibility and virulence in the pediatric population [44]. In Ontario and Québec, the provinces from which most of our study subjects have been reported, the third wave of the pandemic that occurred from March to May 2021 was mostly comprised of the Alpha variant [45]. Reassuringly, we did not find any significant difference in the proportion of infants

**Table 3. Symptoms, laboratory findings, and clinical syndromes of infants with SARS-CoV-2 infection.**

| Clinical features | All Cases | Outpatients | Inpatients | |
| --- | --- | --- | --- | --- |
| | | | **Not COVID-19 related** | **COVID-19 related** |
| **Total cases reported, N** | 531 | 332 | 58 | 141 |
| **COVID-19 severity, n (%)** | | | | |
| Asymptomatic | 66 (12.4) | 22 (6.6) | 44 (75.9) | - - - |
| Outpatient care | 310 (58.4) | 310 (93.4) | - - - | - - - |
| Inpatient care, mild | 125 (23.5) | - - - | 14 (24.1) | 111 (78.7) |
| Inpatient care, moderate | 10 (1.9) | - - - | - - - | 10 (7.1) |
| Inpatient care, severe | 20 (3.8) | - - - | - - - | 20 (14.2) |
| **Symptoms and signs, n (%)** | | | | |
| Fever | 353 (66.5) | 232 (69.9) | 14 (24.1) | 107 (75.9) |
| Coryza | 250 (47.1) | 188 (56.6) | 6 (10.3) | 56 (39.7) |
| Cough | 198 (37.3) | 143–146 (43.1–44.0) | <5 (<8.6) | 50–53 (35.5–37.6) |
| Decreased oral intake | 133 (25.0) | 77 (23.2) | 6 (10.3) | 50 (35.5) |
| Vomiting | 93 (17.5) | 57 (17.2) | 10 (17.2) | 26 (18.4) |
| Diarrhea | 78 (14.7) | 51–54 (15.4–16.3) | <5 (<8.6) | 22–25 (15.6–17.7) |
| Respiratory distress | 55 (10.4) | 17–20 (5.1–6.0) | <5 (<8.6) | 32–35 (22.7–24.8) |
| Lethargy | 53 (10.0) | 19 (5.7) | 7 (12.1) | 27 (19.1) |
| Rash | 49 (9.2) | 35–38 (10.5–11.4) | <5 (<8.6) | 10–13 (7.1–9.2) |
| Irritability[1] | 46 (8.7) | 29–32 (8.7–9.6) | <5 (<8.6) | 12–15 (8.5–10.6) |
| Sneezing | 44 (8.3) | 31 (9.3) | 0 (0.0) | 13 (9.2) |
| Tachycardia[1] | 27 (5.1) | 21 (6.3) | <5 (<8.6) | <5 (<3.5) |
| Conjunctivitis | 13 (2.4) | 6 (1.8) | 0 (0.0) | 7 (5.0) |
| **Laboratory findings, n (%)** | | | | |
| Neutropenia | 34 (6.4) | 9–12 (2.7–3.6) | <5 (<8.6) | 19–22 (13.5–15.6) |
| Lymphopenia | 22 (4.1) | 7 (2.1) | 0 (0.0) | 15 (10.6) |
| Thrombocytosis | 14 (2.6) | 5–8 (1.5–2.4) | <5 (<8.6) | 5–8 (3.5–5.7) |
| Anemia | 8 (1.5) | 0 (0.0) | <5 (<8.6) | 5–7 (3.5–5.0) |
| **Clinical syndromes, n (%)** | | | | |
| Upper respiratory tract infection | 321 (60.5) | 231 (69.6) | 8 (13.8) | 82 (58.2) |
| Gastrointestinal | 161 (30.3) | 103 (31.0) | 13 (22.4) | 45 (31.9) |
| Bronchiolitis | 20 (3.8) | 6 (1.8) | 0 (0.0) | 14 (9.9) |
| Pneumonia | 14 (2.6) | 0 (0.0) | <5 (<8.6) | 10–13 (7.1–9.2) |
| Coagulopathy | 7 (1.3) | 0 (0.0) | <5 (<8.6) | 5–6 (3.5–4.3) |
| Seizure(s) | 7 (1.3) | 0 (0.0) | <5 (<8.6) | <5 (<3.5) |
| Hepatitis | 5 (0.9) | <5 (<1.5) | 0 (0.0) | <5 (<3.5) |
| Acute respiratory distress syndrome | <5 (<0.9) | 0 (0.0) | 0 (0.0) | <5 (<3.5) |
| Cardiac dysfunction | <5 (<0.9) | 0 (0.0) | <5 (<8.6) | <5 (<3.5) |
| **Abnormal CXR, n (%) / cases with imaging** | 32/132 (24.2) | <5/28 (<17.9) | 5–8/19 (26.3–42.1) | 21–24/85 (24.7–28.2) |
| **Any respiratory support required, n (%)** | - - - | - - - | - - - | 23 (16.3) |
| Low-flow oxygen | - - - | - - - | - - - | 15 (10.6) |
| High-flow nasal cannula | - - - | - - - | - - - | 5 (3.5) |
| Non-invasive ventilation | - - - | - - - | - - - | <5 (<3.5) |
| Mechanical ventilation[2] | - - - | - - - | - - - | 5 (3.5) |
| Vasopressors | - - - | - - - | - - - | 0 (0.0) |
| **Admitted to ICU, n (%)** | - - - | - - - | - - - | 14 (9.9) |

*(Continued)*

**Table 3.** (Continued)

| Clinical features | All Cases | Outpatients | Inpatients | |
|---|---|---|---|---|
| | | | **Not COVID-19 related** | **COVID-19 related** |
| **Infant died, n (%)** | <5 (<0.9) | - - - | - - - | <5 (<3.5) |

[1]Symptoms included as write-in notes only and may therefore be underrepresented.

[2]Includes conventional mechanical and high-frequency oscillatory ventilation.

admitted for COVID-19 during the third wave of the pandemic, or between SARS-CoV-2 lineages. However, we have limited data on Delta (B.1.617) variant, and no data on Omicron (B.1.1.529) variant. The first cases of the Delta VOC were identified in Québec on April 21, 2021 by using whole genome sequencing (WGS), as the variant is negative for the detection of N501Y and E484K mutations (widely used for VOC screening) [16]. Since we were unable to obtain data on WGS in our study, SARS-CoV-2 isolates that were negative for both N501Y and E484K mutations from April 21[st] to May 31[st] may have been Delta VOC that were misclassified as wild-type SARS-CoV-2. All data preceded the Omicron wave of the pandemic. Further data on possible increased susceptibility and severe outcomes in infants secondary to the Delta and Omicron VOC will be needed.

This study is also limited by the voluntary nature of reporting in the CPSP system, and therefore, we have not captured data on every infant who tested positive for SARS-CoV-2 in Canada, nor infants who were infected but for whom parents did not seek health care. Under-reporting of outpatient cases may have also caused an information bias in comparison between hospitalized and non-hospitalized patients, likely leading to over-estimation of the proportion of infants who require hospitalization for COVID-19. For cases diagnosed in the first days of life, we were also unable to differentiate SARS-CoV-2 infection acquired congenitally versus postnatally, and the possible impact of the method of transmission on disease severity in infants. Although we described the distribution of cases by population group, we are unable to draw any conclusions because this variable was reported by physicians, and not by families. Lastly, we were unable to obtain information on the vaccination status or prior infection of the infants' mothers.

## Conclusion

Our study has shown that SARS-CoV-2 infection prior to the Omicron wave generally causes mild illness in infants, but that comorbid conditions and younger age were associated with COVID-19-related hospitalization. Thorough understanding of the spectrum of disease and characteristics associated with severe disease in young infants may inform the clinical management and public health interventions targeted to this population. In the absence of available COVID-19 vaccines in infants, increasing acceptance and uptake of COVID-19 vaccines in pregnant women is likely to be an important strategy to protect young infants against SARS-CoV-2 infection.

## Supporting information

**S1 Fig. Odds ratios of COVID-19 admission by continuous infant age, compared to infants at 100 days of age.** Odds ratios are adjusted by infant sex, gestational age category, comorbid conditions, and wave of pandemic. Continuous infant age was analyzed as a quadratic term. (TIF)

**S1 Table. SARS-CoV-2 CPSP severity criteria.**
(PDF)

**S2 Table. Symptoms, laboratory findings, and clinical syndromes of infants with SARS-CoV-2 infection by age group.**
(PDF)

**S3 Table. Logistic regression sensitivity analysis of factors associated with hospital admission due to COVID-19-related disease.**
(PDF)

**S4 Table. Characteristics of infants with SARS-CoV-2 infection reported from the Centre Hospitalier Universitaire Sainte-Justine, Hospital for Sick Children, or Montreal Children's Hospital.**
(PDF)

**S5 Table. Comparison of CPSP and modified Dong severity criteria.**
(PDF)

**S6 Table. Characteristics of admitted infants with severe COVID-19.**
(PDF)

**S7 Table. Treatments administered to infants with COVID-19.**
(PDF)

## Acknowledgments

The authors wish to thank the paediatricians, paediatric sub-specialists, and health professionals who voluntarily respond to CPSP surveys. We also wish to thank Dr. Jim Kellner, MD, MSc for his guidance and support, as well as the members and leadership of the Paediatric Inpatient Research Network (PIRN) for cases reported. We thank Lorraine Piché for her assistance at the Montreal Children's Hospital site. Lastly, the authors also wish to thank members of the CPSP Scientific Steering Committee who oversee the program and the managers and staff of the CPSP who implement the program.

**Complete list of CPSP authors**:

| | |
|---|---|
| Olivier Drouin | Division of General Pediatrics, Department of Pediatrics, CHU Sainte-Justine, Montréal, Que. |
| | Department of Social and Preventive Medicine, School of Public Health, Université de Montréal, Montréal, Que. |
| Charlotte Moore-Hepburn | Division of Paediatric Medicine, The Hospital for Sick Children, Toronto, Ont. |
| Daniel S. Farrar | Centre for Global Child Health, The Hospital for Sick Children, Toronto, Ont. |
| Krista Baerg | Department of Pediatrics, University of Saskatchewan, Saskatoon, Sask. |
| | Jim Pattison Children's Hospital, Saskatchewan Health Authority, Saskatoon, Sask. |
| Kevin Chan | Department of Children's and Women's Health, Trillium Health Partners, Mississauga, Ont. |
| | Department of Pediatrics, University of Toronto, Toronto, Ont. |

(*Continued*)

(Continued)

| | |
|---|---|
| Claude Cyr | Service de Soins Intensifs Pédiatriques, Centre Hospitalier Universitaire de Sherbrooke, Sherbrooke, Que. |
| | Faculté de Médecine, Université de Sherbrooke, Sherbrooke, Que. |
| Elizabeth J. Donner | Division of Neurology, The Hospital for Sick Children, University of Toronto, Toronto, Ont. |
| Joanne E. Embree | Department of Paediatrics and Child Health, University of Manitoba, Winnipeg, Man. |
| | Department of Medical Microbiology and Infectious Diseases, University of Manitoba, Winnipeg, Man. |
| Catherine Farrell | Division of Pediatric Intensive Care, Department of Pediatrics, CHU Sainte-Justine, Montréal, Que. |
| Sarah Forgie | Division of Infectious Diseases, Department of Pediatrics, University of Alberta, Edmonton, Alta. |
| | Stollery Children's Hospital, Edmonton, Alta. |
| Ryan Giroux | Division of Paediatric Medicine, The Hospital for Sick Children, Toronto, Ont. |
| Kristopher T. Kang | Department of Pediatrics, University of British Columbia, Vancouver, BC. |
| Melanie King | Canadian Paediatric Surveillance Program, Canadian Paediatric Society, Ottawa |
| Melanie Laffin | Canadian Paediatric Surveillance Program, Canadian Paediatric Society, Ottawa |
| Julia Orkin | Division of Paediatric Medicine, The Hospital for Sick Children, Toronto, Ont. |
| | Child Health Evaluative Sciences, The Hospital for Sick Children, Toronto, Ont. |
| Jesse Papenburg | Division of Pediatric Infectious Diseases, Department of Pediatrics, Montreal Children's Hospital, Montréal, Que. |
| | Division of Microbiology, Department of Clinical Laboratory Medicine, McGill University Health Centre, Montréal, Que. |
| Catherine M. Pound | Division of General Pediatrics, Department of Pediatrics, Children's Hospital of Eastern Ontario, Ottawa, Ont. |
| Victoria E. Price | Division of Pediatric Hematology/Oncology, Department of Pediatrics, Dalhousie University, Halifax, NS. |
| Rupena Purewal | Department of Pediatrics, University of Saskatchewan, Saskatoon, Sask. |
| | Division of Pediatric Infectious Diseases, Jim Pattison Children's Hospital, Saskatchewan Health Authority, Saskatoon, Sask. |
| Manish Sadarangani | Department of Pediatrics, University of British Columbia, Vancouver, BC. |
| | Vaccine Evaluation Center, BC Children's Hospital Research Institute, Vancouver, BC. |
| Marina I. Salvadori | Public Health Agency of Canada, Ottawa |
| | Division of Pediatric Infectious Diseases, Department of Pediatrics, Montreal Children's Hospital, Montréal, Que. |
| Karina A. Top | Department of Pediatrics, Dalhousie University, Halifax, NS. |
| Isabelle Viel-Thériault | Division of Infectious Diseases, Department of Pediatrics, CHU de Québec-Université Laval, Québec City, Que. |
| Fatima Kakkar | Division of Infectious Diseases, CHU Sainte-Justine, Montréal, Que. |
| **Shaun K. Morris** **Lead author** Shaun.morris@sickkids.ca | Division of Infectious Diseases, The Hospital for Sick Children, Toronto, Ont. |
| | Centre for Global Child Health, The Hospital for Sick Children, Toronto, Ont. |
| | Department of Pediatrics, Temerty Faculty of Medicine, University of Toronto, Toronto, Canada |
| | Clinical Public Health, Dalla Lana School of Public Health, University of Toronto, Toronto, Ont. |

## Author Contributions

**Conceptualization:** Pierre-Philippe Piché-Renaud, Luc Panetta, Charlotte Moore-Hepburn, Olivier Drouin, Jesse Papenburg, Marina I. Salvadori, Fatima Kakkar, Shaun K. Morris.

**Data curation:** Daniel S. Farrar, Melanie Laffin.

**Formal analysis:** Pierre-Philippe Piché-Renaud, Luc Panetta, Daniel S. Farrar, Olivier Drouin, Fatima Kakkar, Shaun K. Morris.

**Funding acquisition:** Charlotte Moore-Hepburn, Olivier Drouin, Jesse Papenburg, Marina I. Salvadori, Fatima Kakkar, Shaun K. Morris.

**Investigation:** Pierre-Philippe Piché-Renaud, Luc Panetta, Daniel S. Farrar, Charlotte Moore-Hepburn, Olivier Drouin, Jesse Papenburg, Marina I. Salvadori, Melanie Laffin, Fatima Kakkar, Shaun K. Morris.

**Methodology:** Pierre-Philippe Piché-Renaud, Luc Panetta, Daniel S. Farrar, Charlotte Moore-Hepburn, Olivier Drouin, Jesse Papenburg, Marina I. Salvadori, Fatima Kakkar, Shaun K. Morris.

**Project administration:** Pierre-Philippe Piché-Renaud, Daniel S. Farrar.

**Resources:** Jesse Papenburg.

**Software:** Daniel S. Farrar.

**Supervision:** Charlotte Moore-Hepburn, Olivier Drouin, Fatima Kakkar, Shaun K. Morris.

**Visualization:** Pierre-Philippe Piché-Renaud, Melanie Laffin.

**Writing – original draft:** Pierre-Philippe Piché-Renaud, Luc Panetta.

**Writing – review & editing:** Daniel S. Farrar, Charlotte Moore-Hepburn, Olivier Drouin, Jesse Papenburg, Marina I. Salvadori, Melanie Laffin, Fatima Kakkar, Shaun K. Morris.

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
