## [Decision Letter · Decision Letter 0]

6 Jun 2022

PONE-D-22-09133Clinical manifestations and disease severity of SARS-CoV-2 infection among infants in CanadaPLOS ONE

Dear Dr. Morris,

Thank you for submitting your manuscript to PLOS ONE. After careful consideration, we feel that it has merit but does not fully meet PLOS ONE’s publication criteria as it currently stands. Therefore, we invite you to submit a revised version of the manuscript that addresses the points raised during the review process.

We look forward to receiving your revised manuscript.

Kind regards,

Everton Falcão de Oliveira, Ph.D

Academic Editor

PLOS ONE

Journal Requirements:

3.Please provide additional details regarding participant consent. In the ethics statement in the Methods and online submission information, please ensure that you have specified (1) whether consent was informed and (2) what type you obtained (for instance, written or verbal, and if verbal, how it was documented and witnessed). If your study included minors, state whether you obtained consent from parents or guardians. If the need for consent was waived by the ethics committee, please include this information.

5. Thank you for stating the following in the Funding Section of your manuscript: 

"OD & FK are supported by a Clinical Research Scholars Award, from the Fonds de recherche du Québec – Santé. We gratefully acknowledge the financial support received from the Public Health Agency of Canada to the Canadian Paediatric Surveillance Program, in support of the COVID-19 study."

"OD & FK are supported by a Clinical Research Scholars Award, from the Fonds de recherche du Québec – Santé. We gratefully acknowledge the financial support received from the Public Health Agency of Canada to the Canadian Paediatric Surveillance Program, in support of the COVID-19 study."

6. Thank you for stating the following in the Competing Interests section: 

"JP reports grants to his institution from MedImmune, Merck, Sanofi Pasteur and AbbVie, and speaker fees from AbbVie and AstraZeneca, all outside of the submitted work. MS is supported via salary awards from the BC Children’s Hospital Foundation, the Canadian Child Health Clinician Scientist Program and the Michael Smith Foundation for Health Research. MS has been an investigator on projects funded by GlaxoSmithKline, Merck, Pfizer, Sanofi-Pasteur, Seqirus, Symvivo and VBI Vaccines. All funds have been paid to his institute, and he has not received any personal payments. PPPR has been co-investigator on an investigator-led project funded by Pfizer that is unrelated to this study. RP is a consultant for Verity Pharmaceuticals. SKM is co-principal investigator on an investigator-led grant from Pfizer, has served on an ad-hoc advisory boards for Pfizer and Sanofi Pasteur, and has received speaker fees from GlaxoSmithKline, all unrelated to this study. "

7. In your Data Availability statement, you have not specified where the minimal data set underlying the results described in your manuscript can be found. PLOS defines a study's minimal data set as the underlying data used to reach the conclusions drawn in the manuscript and any additional data required to replicate the reported study findings in their entirety. All PLOS journals require that the minimal data set be made fully available. For more information about our data policy, please see http://journals.plos.org/plosone/s/data-availability.

8. PLOS requires an ORCID iD for the corresponding author in Editorial Manager on papers submitted after December 6th, 2016. Please ensure that you have an ORCID iD and that it is validated in Editorial Manager. To do this, go to ‘Update my Information’ (in the upper left-hand corner of the main menu), and click on the Fetch/Validate link next to the ORCID field. This will take you to the ORCID site and allow you to create a new iD or authenticate a pre-existing iD in Editorial Manager. Please see the following video for instructions on linking an ORCID iD to your Editorial Manager account: https://www.youtube.com/watch?v=_xcclfuvtxQ

9. One of the noted authors is a group or consortium Canadian Paediatric Surveillance Program COVID-19 Study Team. In addition to naming the author group, please list the individual authors and affiliations within this group in the acknowledgments section of your manuscript. Please also indicate clearly a lead author for this group along with a contact email address.

Reviewers' comments:

Reviewer's Responses to Questions

**Comments to the Author**

1. Is the manuscript technically sound, and do the data support the conclusions?

Reviewer #1: Yes

2. Has the statistical analysis been performed appropriately and rigorously? 

Reviewer #1: I Don't Know

3. Have the authors made all data underlying the findings in their manuscript fully available?

Reviewer #1: Yes

4. Is the manuscript presented in an intelligible fashion and written in standard English?

Reviewer #1: Yes

5. Review Comments to the Author

Reviewer #1: This is a well written paper that provides a nice description of infants and their manifestations from COVID-19. I only have a two major thread of comments. The issue is the selection approach is not clear. The results from the selection approach are biased as well (connected to the patient selection approach), and this need to be clarified. These comments are specifically outlined below.

Abstract

1. Page 3, line 69: 37.5% admission clearly is biased if you are looking at this as all infants that acquired COVID-19. I think its important to make this clear throughout the paper. For instance on page 4, line 78, the authors state that most are mildly ill, but this doesn't compute with the 35% admission rate. The denominator of asymptomatic and not sick infants is not part of the study, so you need to be clear about this.

METHODS

2. Page 6, line 109: The CPSP needs to be clarified. Are these outpatient clinics? How does the hospital enrollment overlap or not?

3. Clarify that you did not include asymptomatic and those without significant disease from the CPSP, since it appears you did not. Or if I am not understanding this, please clarify.

4. Page 7, lines 136 - 139: It appears you have included 2800 outpatient practices and then also added 3 hospitals. Are the same participants included in both? If so, was that to obtain the outpatient + inpatient course together? Or are these separate? The three categories appears to me to be (1) CPSP, (2) either CPSP or hospital, (3) hospitals. I am not sure if this is right and it matters since it impacts your results.

RESULTS

5. Clarify the results with the selection criteria so its more clear how the results are connected (or not).

6. Page 15, lines 253 - 256 -- The criteria for hospitalization in young infants is pretty large -- fever in particular. While this is noted in the Conclusions, the results seem driven by this admission criteria and it should be pointed out sooner.

7. Page 15, line 257 -- Please also give the age in months since it seems odd to flip from one to the other.

8. Page 19, line 280: The fact that 78.7% of admitted children had mild disease shows the bias based on the selection approach and also the admission criteria, and should not be considered a finding of how sick these children are.

DISCUSSION

9. Page 23, second paragraph -- I do not believe the authors showed that COVID-19 is uncommon in children with this paper.

6. PLOS authors have the option to publish the peer review history of their article (what does this mean?). If published, this will include your full peer review and any attached files.

Reviewer #1: No

---

## [Author Response · Author response to Decision Letter 0]

20 Jul 2022

Reviewer #1: This is a well written paper that provides a nice description of infants and their manifestations from COVID-19. I only have a two major thread of comments. The issue is the selection approach is not clear. The results from the selection approach are biased as well (connected to the patient selection approach), and this need to be clarified. These comments are specifically outlined below.

We with to thank the reviewer for the comments and feedback on our manuscript. Below are our responses to the detailed comments. We are hoping that the reviewer will find that the updated manuscript has appropriately answered all of their queries.

Abstract

1. Page 3, line 69: 37.5% admission clearly is biased if you are looking at this as all infants that acquired COVID-19. I think its important to make this clear throughout the paper. For instance on page 4, line 78, the authors state that most are mildly ill, but this doesn't compute with the 35% admission rate. The denominator of asymptomatic and not sick infants is not part of the study, so you need to be clear about this.

The abstract conclusion was modified to highlight that severe COVID-19 was found to be rare (even amongst hospitalized cases) and the wording on mild disease was removed. Cases of asymptomatic SARS-CoV-2 infection could be included in the study, however, there may indeed be selection bias and under-reporting as described in the study limitation. 

METHODS

2. Page 6, line 109: The CPSP needs to be clarified. Are these outpatient clinics? How does the hospital enrollment overlap or not?

Pediatricians are members of the CPSP independent of their practice location, may it be in working in hospital settings or outpatient clinics. The pediatricians themselves form the surveillance network rather than the clinics or hospitals. This was clarified in the methods. 

3. Clarify that you did not include asymptomatic and those without significant disease from the CPSP, since it appears you did not. Or if I am not understanding this, please clarify.

Asymptomatic cases could be reported and were classified separately under the severity classification (for example, as reported in Table 3). This was clarified in the methods and is also reported in the Study definitions section (under the definition for asymptomatic cases). 

4. Page 7, lines 136 - 139: It appears you have included 2800 outpatient practices and then also added 3 hospitals. Are the same participants included in both? If so, was that to obtain the outpatient + inpatient course together? Or are these separate? The three categories appears to me to be (1) CPSP, (2) either CPSP or hospital, (3) hospitals. I am not sure if this is right and it matters since it impacts your results.

The CPSP is a network of 2,800 pediatricians who are working in both outpatient and hospital settings. The three centers that were mentioned are part of the CPSP in that CPSP pediatricians are working in these institutions, including the main study investigators. The difference with other settings was the sampling strategy of active case finding using EMR system to systematically collect data on infants, which was not implemented in other settings. Cases could only be reported once, and duplicate data was removed. This was specified in the manuscript. 

RESULTS

5. Clarify the results with the selection criteria so its more clear how the results are connected (or not).

As detailed in the methods, all cases of SARS-CoV-2 in infants aged less than one year old who were seen by pediatricians from the CPSP were eligible for inclusion, if they were reported to the network. All microbiologically-confirmed cases of SARS-CoV-2 that were reported to CPSP were included in the analyses. 

6. Page 15, lines 253 - 256 -- The criteria for hospitalization in young infants is pretty large -- fever in particular. While this is noted in the Conclusions, the results seem driven by this admission criteria and it should be pointed out sooner.

A line to that effect was added to the manuscript, highlighting the percentage of infants <1 month of age who experienced fever. 

7. Page 15, line 257 -- Please also give the age in months since it seems odd to flip from one to the other.

Age was modified from days to months throughout the manuscript.

8. Page 19, line 280: The fact that 78.7% of admitted children had mild disease shows the bias based on the selection approach and also the admission criteria, and should not be considered a finding of how sick these children are. 

As detailed in the methods, all cases of SARS-CoV-2 in infants aged less than one year old who were reported to CPSP were included, without further selection of cases. The study aimed to capture all cases that met the inclusion criteria, as detailed in the methods section. Criteria for hospital admission were entirely clinical, made by clinicians, and there were no study-directed decisions or criteria about hospital admission. The reported percentage of 78.7% represents the percentage of all infants admitted for COVID-19 who had mild disease, based on the pre-established criteria for severity. 

DISCUSSION

9. Page 23, second paragraph -- I do not believe the authors showed that COVID-19 is uncommon in children with this paper.

We agree with the reviewer that the study has not shown that COVID-19 was uncommon in infants. This paragraph states that severe COVID-19 was uncommon within all cases that have been reported (3.8%).

---

## [Editor Report · Decision Letter 1]

25 Jul 2022

Clinical manifestations and disease severity of SARS-CoV-2 infection among infants in Canada

PONE-D-22-09133R1

Dear Dr. Morris,

We’re pleased to inform you that your manuscript has been judged scientifically suitable for publication and will be formally accepted for publication once it meets all outstanding technical requirements.

Kind regards,

Everton Falcão de Oliveira, Ph.D

Academic Editor

PLOS ONE
---

## [Editor Report · Acceptance letter]

2 Aug 2022

PONE-D-22-09133R1 

Clinical manifestations and disease severity of SARS-CoV-2 infection among infants in Canada 

Dear Dr. Morris:

I'm pleased to inform you that your manuscript has been deemed suitable for publication in PLOS ONE. Congratulations! Your manuscript is now with our production department. 

Kind regards, 

on behalf of

Dr. Everton Falcão de Oliveira 

Academic Editor

PLOS ONE